# Triangle Generative Adversarial Networks

**Zhe Gan**$^*$, **Liqun Chen**$^*$, **Weiyao Wang, Yunchen Pu, Yizhe Zhang,**
**Hao Liu, Chunyuan Li, Lawrence Carin**
Duke University
zhe.gan@duke.edu

## Abstract

A Triangle Generative Adversarial Network ($\Delta$-GAN) is developed for semi-supervised cross-domain joint distribution matching, where the training data consists of samples from each domain, and supervision of domain correspondence is provided by only a few *paired* samples. $\Delta$-GAN consists of four neural networks, two generators and two discriminators. The generators are designed to learn the two-way conditional distributions between the two domains, while the discriminators implicitly define a ternary discriminative function, which is trained to distinguish real data pairs and two kinds of fake data pairs. The generators and discriminators are trained together using adversarial learning. Under mild assumptions, in theory the joint distributions characterized by the two generators concentrate to the data distribution. In experiments, three different kinds of domain pairs are considered, image-label, image-image and image-attribute pairs. Experiments on semi-supervised image classification, image-to-image translation and attribute-based image generation demonstrate the superiority of the proposed approach.

## 1 Introduction

Generative adversarial networks (GANs) [1] have emerged as a powerful framework for learning generative models of arbitrarily complex data distributions. When trained on datasets of natural images, significant progress has been made on generating realistic and sharp-looking images [2, 3]. The original GAN formulation was designed to learn the data distribution in one domain. In practice, one may also be interested in matching two joint distributions. This is an important task, since mapping data samples from one domain to another has a wide range of applications. For instance, matching the joint distribution of image-text pairs allows simultaneous image captioning and text-conditional image generation [4], while image-to-image translation [5] is another challenging problem that requires matching the joint distribution of image-image pairs.

In this work, we are interested in designing a GAN framework to match joint distributions. If paired data are available, a simple approach to achieve this is to train a conditional GAN model [4, 6], from which a joint distribution is readily manifested and can be matched to the empirical joint distribution provided by the paired data. However, fully supervised data are often difficult to acquire. Several methods have been proposed to achieve unsupervised joint distribution matching without any paired data, including DiscoGAN [7], CycleGAN [8] and DualGAN [9]. Adversarially Learned Inference (ALI) [10] and Bidirectional GAN (BiGAN) [11] can be readily adapted to this case as well. Though empirically achieving great success, in principle, there exist infinitely many possible mapping functions that satisfy the requirement to map a sample from one domain to another. In order to alleviate this nonidentifiability issue, paired data are needed to provide proper supervision to inform the model the kind of joint distributions that are desired.

This motivates the proposed Triangle Generative Adversarial Network ($\Delta$-GAN), a GAN framework that allows semi-supervised joint distribution matching, where the supervision of domain

---

$^*$ Equal contribution.

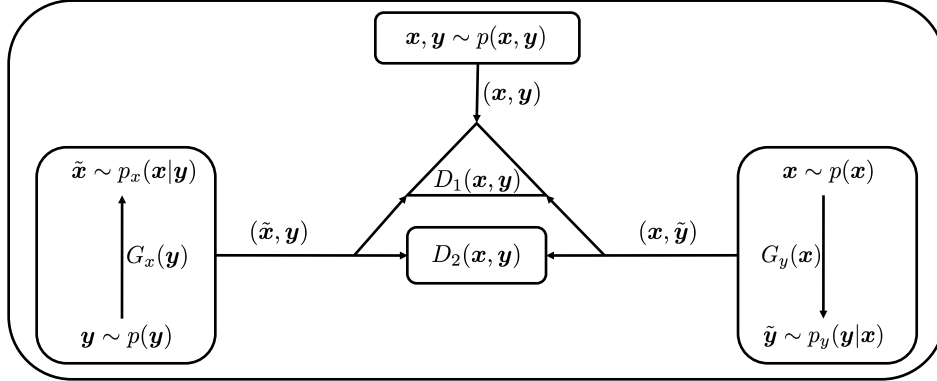

Figure 1: Illustration of the Triangle Generative Adversarial Network ($\Delta$-GAN).

correspondence is provided by a few paired samples. $\Delta$-GAN consists of two generators and two discriminators. The generators are designed to learn the bidirectional mappings between domains, while the discriminators are trained to distinguish real data pairs and two kinds of fake data pairs. Both the generators and discriminators are trained together via adversarial learning.

$\Delta$-GAN bears close resemblance to Triple GAN [12], a recently proposed method that can also be utilized for semi-supervised joint distribution mapping. However, there exist several key differences that make our work unique. First, $\Delta$-GAN uses two discriminators in total, which implicitly defines a ternary discriminative function, instead of a binary discriminator as used in Triple GAN. Second, $\Delta$-GAN can be considered as a combination of conditional GAN and ALI, while Triple GAN consists of two conditional GANs. Third, the distributions characterized by the two generators in both $\Delta$-GAN and Triple GAN concentrate to the data distribution in theory. However, when the discriminator is optimal, the objective of $\Delta$-GAN becomes the Jensen-Shannon divergence (JSD) among three distributions, which is symmetric; the objective of Triple GAN consists of a JSD term plus a Kullback-Leibler (KL) divergence term. The asymmetry of the KL term makes Triple GAN more prone to generating fake-looking samples [13]. Lastly, the calculation of the additional KL term in Triple GAN is equivalent to calculating a supervised loss, which requires the explicit density form of the conditional distributions, which may not be desirable. On the other hand, $\Delta$-GAN is a fully adversarial approach that does not require that the conditional densities can be computed; $\Delta$-GAN only require that the conditional densities can be sampled from in a way that allows gradient backpropagation.

$\Delta$-GAN is a general framework, and can be used to match any joint distributions. In experiments, in order to demonstrate the versatility of the proposed model, we consider three domain pairs: image-label, image-image and image-attribute pairs, and use them for semi-supervised classification, image-to-image translation and attribute-based image editing, respectively. In order to demonstrate the scalability of the model to large and complex datasets, we also present attribute-conditional image generation on the COCO dataset [14].

## 2 Model

### 2.1 Generative Adversarial Networks (GANs)

Generative Adversarial Networks (GANs) [1] consist of a generator $G$ and a discriminator $D$ that compete in a two-player minimax game, where the generator is learned to map samples from an arbitray latent distribution to data, while the discriminator tries to distinguish between real and generated samples. The goal of the generator is to "fool" the discriminator by producing samples that are as close to real data as possible. Specifically, $D$ and $G$ are learned as

$$\min_{G} \max_{D} V(D, G) = \mathbb{E}_{\boldsymbol{x} \sim p(\boldsymbol{x})}[\log D(\boldsymbol{x})] + \mathbb{E}_{\boldsymbol{z} \sim p_z(\boldsymbol{z})}[\log(1 - D(G(\boldsymbol{z})))] , \qquad (1)$$

where $p(\boldsymbol{x})$ is the true data distribution, and $p_z(\boldsymbol{z})$ is usually defined to be a simple distribution, such as the standard normal distribution. The generator $G$ implicitly defines a probability distribution $p_g(\boldsymbol{x})$ as the distribution of the samples $G(\boldsymbol{z})$ obtained when $\boldsymbol{z} \sim p_z(\boldsymbol{z})$. For any fixed generator

$G$, the optimal discriminator is $D(\boldsymbol{x}) = \frac{p(\boldsymbol{x})}{p_g(\boldsymbol{x})+p(\boldsymbol{x})}$. When the discriminator is optimal, solving this adversarial game is equivalent to minimizing the Jenson-Shannon Divergence (JSD) between $p(\boldsymbol{x})$ and $p_g(\boldsymbol{x})$ [1]. The global equilibrium is achieved if and only if $p(\boldsymbol{x}) = p_g(\boldsymbol{x})$.

## 2.2 Triangle Generative Adversarial Networks ($\Delta$-GANs)

We now extend GAN to $\Delta$-GAN for joint distribution matching. We first consider $\Delta$-GAN in the supervised setting, and then discuss semi-supervised learning in Section 2.4. Consider two related domains, with $\boldsymbol{x}$ and $\boldsymbol{y}$ being the data samples for each domain. We have fully-paired data samples that are characterized by the joint distribution $p(\boldsymbol{x}, \boldsymbol{y})$, which also implies that samples from both the marginal $p(\boldsymbol{x})$ and $p(\boldsymbol{y})$ can be easily obtained.

$\Delta$-GAN consists of two generators: (*i*) a generator $G_x(\boldsymbol{y})$ that defines the conditional distribution $p_x(\boldsymbol{x}|\boldsymbol{y})$, and (*ii*) a generator $G_y(\boldsymbol{x})$ that characterizes the conditional distribution in the other direction $p_y(\boldsymbol{y}|\boldsymbol{x})$. $G_x(\boldsymbol{y})$ and $G_y(\boldsymbol{x})$ may also implicitly contain a random latent variable $\boldsymbol{z}$ as input, *i.e.*, $G_x(\boldsymbol{y}, \boldsymbol{z})$ and $G_y(\boldsymbol{x}, \boldsymbol{z})$. In the $\Delta$-GAN game, after a sample $\boldsymbol{x}$ is drawn from $p(\boldsymbol{x})$, the generator $G_y$ produces a pseudo sample $\tilde{\boldsymbol{y}}$ following the conditional distribution $p_y(\boldsymbol{y}|\boldsymbol{x})$. Hence, the fake data pair $(\boldsymbol{x}, \tilde{\boldsymbol{y}})$ is a sample from the joint distribution $p_y(\boldsymbol{x}, \boldsymbol{y}) = p_y(\boldsymbol{y}|\boldsymbol{x})p(\boldsymbol{x})$. Similarly, a fake data pair $(\tilde{\boldsymbol{x}}, \boldsymbol{y})$ can be sampled from the generator $G_x$ by first drawing $\boldsymbol{y}$ from $p(\boldsymbol{y})$ and then drawing $\tilde{\boldsymbol{x}}$ from $p_x(\boldsymbol{x}|\boldsymbol{y})$; hence $(\tilde{\boldsymbol{x}}, \boldsymbol{y})$ is sampled from the joint distribution $p_x(\boldsymbol{x}, \boldsymbol{y}) = p_x(\boldsymbol{x}|\boldsymbol{y})p(\boldsymbol{y})$. As such, the generative process between $p_x(\boldsymbol{x}, \boldsymbol{y})$ and $p_y(\boldsymbol{x}, \boldsymbol{y})$ is reversed.

The objective of $\Delta$-GAN is to match the three joint distributions: $p(\boldsymbol{x}, \boldsymbol{y})$, $p_x(\boldsymbol{x}, \boldsymbol{y})$ and $p_y(\boldsymbol{x}, \boldsymbol{y})$. If this is achieved, we are ensured that we have learned a bidirectional mapping $p_x(\boldsymbol{x}|\boldsymbol{y})$ and $p_y(\boldsymbol{y}|\boldsymbol{x})$ that guarantees the generated fake data pairs $(\tilde{\boldsymbol{x}}, \boldsymbol{y})$ and $(\boldsymbol{x}, \tilde{\boldsymbol{y}})$ are indistinguishable from the true data pairs $(\boldsymbol{x}, \boldsymbol{y})$. In order to match the joint distributions, an adversarial game is played. Joint pairs are drawn from three distributions: $p(\boldsymbol{x}, \boldsymbol{y})$, $p_x(\boldsymbol{x}, \boldsymbol{y})$ or $p_y(\boldsymbol{x}, \boldsymbol{y})$, and two discriminator networks are learned to discriminate among the three, while the two generator networks are trained to fool the discriminators.

The value function describing the game is given by

$$\min_{G_x, G_y} \max_{D_1, D_2} V(G_x, G_y, D_1, D_2) = \mathbb{E}_{(\boldsymbol{x}, \boldsymbol{y}) \sim p(\boldsymbol{x}, \boldsymbol{y})}[\log D_1(\boldsymbol{x}, \boldsymbol{y})] \tag{2}$$

$$+ \mathbb{E}_{\boldsymbol{y} \sim p(\boldsymbol{y}), \tilde{\boldsymbol{x}} \sim p_x(\boldsymbol{x}|\boldsymbol{y})} \Big[ \log \Big( (1 - D_1(\tilde{\boldsymbol{x}}, \boldsymbol{y})) \cdot D_2(\tilde{\boldsymbol{x}}, \boldsymbol{y}) \Big) \Big]$$

$$+ \mathbb{E}_{\boldsymbol{x} \sim p(\boldsymbol{x}), \tilde{\boldsymbol{y}} \sim p_y(\boldsymbol{y}|\boldsymbol{x})} \Big[ \log \Big( (1 - D_1(\boldsymbol{x}, \tilde{\boldsymbol{y}})) \cdot (1 - D_2(\boldsymbol{x}, \tilde{\boldsymbol{y}})) \Big) \Big].$$

The discriminator $D_1$ is used to distinguish whether a sample pair is from $p(\boldsymbol{x}, \boldsymbol{y})$ or not, if this sample pair is not from $p(\boldsymbol{x}, \boldsymbol{y})$, another discriminator $D_2$ is used to distinguish whether this sample pair is from $p_x(\boldsymbol{x}, \boldsymbol{y})$ or $p_y(\boldsymbol{x}, \boldsymbol{y})$. $D_1$ and $D_2$ work cooperatively, and the use of both implicitly defines a ternary discriminative function $D$ that distinguish sample pairs in three ways. See Figure 1 for an illustration of the adversarial game and Appendix B for an algorithmic description of the training procedure.

## 2.3 Theoretical analysis

$\Delta$-GAN shares many of the theoretical properties of GANs [1]. We first consider the optimal discriminators $D_1$ and $D_2$ for any given generator $G_x$ and $G_y$. These optimal discriminators then allow reformulation of objective (2), which reduces to the Jensen-Shannon divergence among the joint distribution $p(\boldsymbol{x}, \boldsymbol{y}), p_x(\boldsymbol{x}, \boldsymbol{y})$ and $p_y(\boldsymbol{x}, \boldsymbol{y})$.

**Proposition 1.** *For any fixed generator $G_x$ and $G_y$, the optimal discriminator $D_1$ and $D_2$ of the game defined by $V(G_x, G_y, D_1, D_2)$ is*

$$D_1^*(\boldsymbol{x}, \boldsymbol{y}) = \frac{p(\boldsymbol{x}, \boldsymbol{y})}{p(\boldsymbol{x}, \boldsymbol{y}) + p_x(\boldsymbol{x}, \boldsymbol{y}) + p_y(\boldsymbol{x}, \boldsymbol{y})}, \quad D_2^*(\boldsymbol{x}, \boldsymbol{y}) = \frac{p_x(\boldsymbol{x}, \boldsymbol{y})}{p_x(\boldsymbol{x}, \boldsymbol{y}) + p_y(\boldsymbol{x}, \boldsymbol{y})} .$$

*Proof.* The proof is a straightforward extension of the proof in [1]. See Appendix A for details. □

**Proposition 2.** *The equilibrium of $V(G_x, G_y, D_1, D_2)$ is achieved if and only if $p(\boldsymbol{x}, \boldsymbol{y}) = p_x(\boldsymbol{x}, \boldsymbol{y}) = p_y(\boldsymbol{x}, \boldsymbol{y})$ with $D_1^*(\boldsymbol{x}, \boldsymbol{y}) = \frac{1}{3}$ and $D_2^*(\boldsymbol{x}, \boldsymbol{y}) = \frac{1}{2}$, and the optimum value is $-3 \log 3$.*

*Proof.* Given the optimal $D_1^*(\boldsymbol{x}, \boldsymbol{y})$ and $D_2^*(\boldsymbol{x}, \boldsymbol{y})$, the minimax game can be reformulated as:

$$C(G_x, G_y) = \max_{D_1, D_2} V(G_x, G_y, D_1, D_2) \tag{3}$$

$$= -3\log 3 + 3 \cdot JSD\Big(p(\boldsymbol{x}, \boldsymbol{y}), p_x(\boldsymbol{x}, \boldsymbol{y}), p_y(\boldsymbol{x}, \boldsymbol{y})\Big) \geq -3\log 3, \tag{4}$$

where $JSD$ denotes the Jensen-Shannon divergence (JSD) among three distributions. See Appendix A for details. □

Since $p(\boldsymbol{x}, \boldsymbol{y}) = p_x(\boldsymbol{x}, \boldsymbol{y}) = p_y(\boldsymbol{x}, \boldsymbol{y})$ can be achieved in theory, it can be readily seen that the learned conditional generators can reveal the true conditional distributions underlying the data, *i.e.*, $p_x(\boldsymbol{x}|\boldsymbol{y}) = p(\boldsymbol{x}|\boldsymbol{y})$ and $p_y(\boldsymbol{y}|\boldsymbol{x}) = p(\boldsymbol{y}|\boldsymbol{x})$.

### 2.4 Semi-supervised learning

In order to further understand $\Delta$-GAN, we write (2) as

$$V = \underbrace{\mathbb{E}_{p(\boldsymbol{x}, \boldsymbol{y})}[\log D_1(\boldsymbol{x}, \boldsymbol{y})] + \mathbb{E}_{p_x(\tilde{\boldsymbol{x}}, \boldsymbol{y})}[\log(1 - D_1(\tilde{\boldsymbol{x}}, \boldsymbol{y}))] + \mathbb{E}_{p_y(\boldsymbol{x}, \tilde{\boldsymbol{y}})}[\log(1 - D_1(\boldsymbol{x}, \tilde{\boldsymbol{y}}))]}_{\text{conditional GAN}} \tag{5}$$

$$+ \underbrace{\mathbb{E}_{p_x(\tilde{\boldsymbol{x}}, \boldsymbol{y})}[\log D_2(\tilde{\boldsymbol{x}}, \boldsymbol{y})] + \mathbb{E}_{p_y(\boldsymbol{x}, \tilde{\boldsymbol{y}})}[\log(1 - D_2(\boldsymbol{x}, \tilde{\boldsymbol{y}}))]}_{\text{BiGAN/ALI}} . \tag{6}$$

The objective of $\Delta$-GAN is a combination of the objectives of conditional GAN and BiGAN. The BiGAN part matches two joint distributions: $p_x(\boldsymbol{x}, \boldsymbol{y})$ and $p_y(\boldsymbol{x}, \boldsymbol{y})$, while the conditional GAN part provides the supervision signal to notify the BiGAN part what joint distribution to match. Therefore, $\Delta$-GAN provides a natural way to perform semi-supervised learning, since the conditional GAN part and the BiGAN part can be used to account for paired and unpaired data, respectively.

However, when doing semi-supervised learning, there is also one potential problem that we need to be cautious about. The theoretical analysis in Section 2.3 is based on the assumption that the dataset is fully supervised, *i.e.*, we have the ground-truth joint distribution $p(\boldsymbol{x}, \boldsymbol{y})$ and marginal distributions $p(\boldsymbol{x})$ and $p(\boldsymbol{y})$. In the semi-supervised setting, $p(\boldsymbol{x})$ and $p(\boldsymbol{y})$ are still available but $p(\boldsymbol{x}, \boldsymbol{y})$ is not. We can only obtain the joint distribution $p_l(\boldsymbol{x}, \boldsymbol{y})$ characterized by the few paired data samples. Hence, in the semi-supervised setting, $p_x(\boldsymbol{x}, \boldsymbol{y})$ and $p_y(\boldsymbol{x}, \boldsymbol{y})$ will try to concentrate to the empirical distribution $p_l(\boldsymbol{x}, \boldsymbol{y})$. We make the assumption that $p_l(\boldsymbol{x}, \boldsymbol{y}) \approx p(\boldsymbol{x}, \boldsymbol{y})$, *i.e.*, the paired data can roughly characterize the whole dataset. For example, in the semi-supervised classification problem, one usually strives to make sure that labels are equally distributed among the labeled dataset.

### 2.5 Relation to Triple GAN

$\Delta$-GAN is closely related to Triple GAN [12]. Below we review Triple GAN and then discuss the main differences. The value function of Triple GAN is defined as follows:

$$V = \mathbb{E}_{p(\boldsymbol{x}, \boldsymbol{y})}[\log D(\boldsymbol{x}, \boldsymbol{y})] + (1 - \alpha)\mathbb{E}_{p_x(\tilde{\boldsymbol{x}}, \boldsymbol{y})}[\log(1 - D(\tilde{\boldsymbol{x}}, \boldsymbol{y}))] + \alpha \mathbb{E}_{p_y(\boldsymbol{x}, \tilde{\boldsymbol{y}})}[\log(1 - D(\boldsymbol{x}, \tilde{\boldsymbol{y}}))]$$
$$+ \mathbb{E}_{p(\boldsymbol{x}, \boldsymbol{y})}[-\log p_y(\boldsymbol{y}|\boldsymbol{x})] , \tag{7}$$

where $\alpha \in (0, 1)$ is a contant that controls the relative importance of the two generators. Let Triple GAN-s denote a simplified Triple GAN model with only the first three terms. As can be seen, Triple GAN-s can be considered as a combination of two conditional GANs, with the importance of each condtional GAN weighted by $\alpha$. It can be proven that Triple GAN-s achieves equilibrium if and only if $p(\boldsymbol{x}, \boldsymbol{y}) = (1 - \alpha)p_x(\boldsymbol{x}, \boldsymbol{y}) + \alpha p_y(\boldsymbol{x}, \boldsymbol{y})$, which is not desirable. To address this problem, in Triple GAN a standard supervised loss $\mathcal{R}_\mathcal{L} = \mathbb{E}_{p(\boldsymbol{x}, \boldsymbol{y})}[-\log p_y(\boldsymbol{y}|\boldsymbol{x})]$ is added. As a result, when the discriminator is optimal, the cost function in Triple GAN becomes:

$$2JSD\Big(p(\boldsymbol{x}, \boldsymbol{y})||((1 - \alpha)p_x(\boldsymbol{x}, \boldsymbol{y}) + \alpha p_y(\boldsymbol{x}, \boldsymbol{y}))\Big) + KL(p(\boldsymbol{x}, \boldsymbol{y})||p_y(\boldsymbol{x}, \boldsymbol{y})) + \text{const.} \tag{8}$$

This cost function has the good property that it has a unique minimum at $p(\boldsymbol{x}, \boldsymbol{y}) = p_x(\boldsymbol{x}, \boldsymbol{y}) = p_y(\boldsymbol{x}, \boldsymbol{y})$. However, the objective becomes asymmetrical. The second KL term pays low cost for generating fake-looking samples [13]. By contrast $\Delta$-GAN directly optimizes the *symmetric* Jensen-Shannon divergence among three distributions. More importantly, the calculation of

$\mathbb{E}_{p(\boldsymbol{x},\boldsymbol{y})}[-\log p_y(\boldsymbol{y}|\boldsymbol{x})]$ in Triple GAN also implies that the explicit density form of $p_y(\boldsymbol{y}|\boldsymbol{x})$ should be provided, which may not be desirable. On the other hand, $\Delta$-GAN only requires that $p_y(\boldsymbol{y}|\boldsymbol{x})$ can be sampled from. For example, if we assume $p_y(\boldsymbol{y}|\boldsymbol{x}) = \int \delta(\boldsymbol{y} - G_y(\boldsymbol{x},\boldsymbol{z}))p(\boldsymbol{z})d\boldsymbol{z}$, and $\delta(\cdot)$ is the Dirac delta function, we can sample $\boldsymbol{y}$ through sampling $\boldsymbol{z}$, however, the density function of $p_y(\boldsymbol{y}|\boldsymbol{x})$ is not explicitly available.

### 2.6 Applications

$\Delta$-GAN is a general framework that can be used for any joint distribution matching. Besides the semi-supervised image classification task considered in [12], we also conduct experiments on image-to-image translation and attribute-conditional image generation. When modeling image pairs, both $p_x(\boldsymbol{x}|\boldsymbol{y})$ and $p_y(\boldsymbol{y}|\boldsymbol{x})$ are implemented without introducing additional latent variables, *i.e.*, $p_x(\boldsymbol{x}|\boldsymbol{y}) = \delta(\boldsymbol{x} - G_x(\boldsymbol{y})), p_y(\boldsymbol{y}|\boldsymbol{x}) = \delta(\boldsymbol{y} - G_y(\boldsymbol{x}))$.

A different strategy is adopted when modeling the image-label/attribute pairs. Specifically, let $\boldsymbol{x}$ denote samples in the image domain, $\boldsymbol{y}$ denote samples in the label/attribute domain. $\boldsymbol{y}$ is a one-hot vector or a binary vector when representing labels and attributes, respectively. When modeling $p_x(\boldsymbol{x}|\boldsymbol{y})$, we assume that $\boldsymbol{x}$ is transformed by the latent style variables $\boldsymbol{z}$ given the label or attribute vector $\boldsymbol{y}$, *i.e.*, $p_x(\boldsymbol{x}|\boldsymbol{y}) = \int \delta(\boldsymbol{x} - G_x(\boldsymbol{y},\boldsymbol{z}))p(\boldsymbol{z})d\boldsymbol{z}$, where $p(\boldsymbol{z})$ is chosen to be a simple distribution (*e.g.*, uniform or standard normal). When learning $p_y(\boldsymbol{y}|\boldsymbol{x})$, $p_y(\boldsymbol{y}|\boldsymbol{x})$ is assumed to be a standard multi-class or multi-label classfier without latent variables $\boldsymbol{z}$. In order to allow the training signal backpropagated from $D_1$ and $D_2$ to $G_y$, we adopt the REINFORCE algorithm as in [12], and use the label with the maximum probability to approximate the expectation over $\boldsymbol{y}$, or use the output of the sigmoid function as the predicted attribute vector.

## 3   Related work

The proposed framework focuses on designing GAN for joint-distribution matching. Conditional GAN can be used for this task if supervised data is available. Various conditional GANs have been proposed to condition the image generation on class labels [6], attributes [15], texts [4, 16] and images [5, 17]. Unsupervised learning methods have also been developed for this task. BiGAN [11] and ALI [10] proposed a method to jointly learn a generation network and an inference network via adversarial learning. Though originally designed for learning the two-way transition between the stochastic latent variables and real data samples, BiGAN and ALI can be directly adapted to learn the joint distribution of two real domains. Another method is called DiscoGAN [7], in which two generators are used to model the bidirectional mapping between domains, and another two discriminators are used to decide whether a generated sample is fake or not in each individual domain. Further, additional reconstructon losses are introduced to make the two generators strongly coupled and also alleviate the problem of mode collapsing. Similiar work includes CycleGAN [8], DualGAN [9] and DTN [18]. Additional weight-sharing constraints are introduced in CoGAN [19] and UNIT [20].

Our work differs from the above work in that we aim at semi-supervised joint distribution matching. The only work that we are aware of that also achieves this goal is Triple GAN. However, our model is distinct from Triple GAN in important ways (see Section 2.5). Further, Triple GAN only focuses on image classification, while $\Delta$-GAN has been shown to be applicable to a wide range of applications.

Various methods and model architectures have been proposed to improve and stabilize the training of GAN, such as feature matching [21, 22, 23], Wasserstein GAN [24], energy-based GAN [25], and unrolled GAN [26] among many other related works. Our work is orthogonal to these methods, which could also be used to improve the training of $\Delta$-GAN. Instead of using adversarial loss, there also exists work that uses supervised learning [27] for joint-distribution matching, and variational autoencoders for semi-supervised learning [28, 29]. Lastly, our work is also closely related to the recent work of [30, 31, 32], which treats one of the domains as latent variables.

## 4   Experiments

We present results on three tasks: (*i*) semi-supervised classification on CIFAR10 [33]; (*ii*) image-to-image translation on MNIST [34] and the edges2shoes dataset [5]; and (*iii*) attribute-to-image generation on CelebA [35] and COCO [14]. We also conduct a toy data experiment to further demonstrate the differences between $\Delta$-GAN and Triple GAN. We implement $\Delta$-GAN without introducing additional regularization unless explicitly stated. All the network architectures are provided in the Appendix.

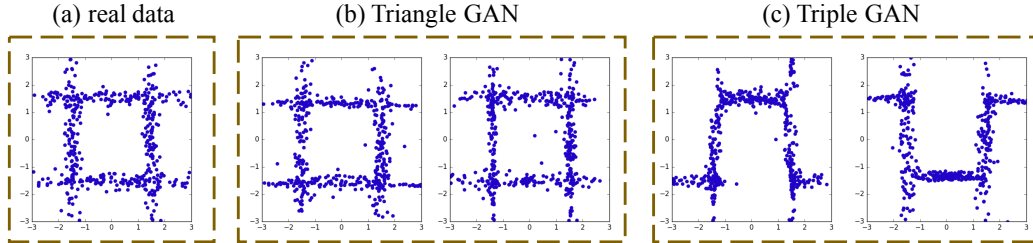

Figure 2: Toy data experiment on $\Delta$-GAN and Triple GAN. (a) the joint distribution $p(x, y)$ of real data. For (b) and (c), the left and right figure is the learned joint distribution $p_x(x, y)$ and $p_y(x, y)$, respectively.

Table 1: Error rates (%) on the partially labeled CIFAR10 dataset.

| Algorithm | $n = 4000$ |
|---|---|
| CatGAN [36] | $19.58 \pm 0.58$ |
| Improved GAN [21] | $18.63 \pm 2.32$ |
| ALI [10] | $17.99 \pm 1.62$ |
| Triple GAN [12] | $16.99 \pm 0.36$ |
| $\Delta$-GAN (ours) | $\mathbf{16.80 \pm 0.42}$ |

Table 2: Classification accuracy (%) on the MNIST-to-MNIST-transpose dataset.

| Algorithm | $n = 100$ | $n = 1000$ | All |
|---|---|---|---|
| DiscoGAN | – | – | $15.00 \pm 0.20$ |
| Triple GAN | $63.79 \pm 0.85$ | $84.93 \pm 1.63$ | $86.70 \pm 1.52$ |
| $\Delta$-GAN | $\mathbf{83.20 \pm 1.88}$ | $\mathbf{88.98 \pm 1.50}$ | $\mathbf{93.34 \pm 1.46}$ |

## 4.1 Toy data experiment

We first compare our method with Triple GAN on a toy dataset. We synthesize data by drawing $(x, y) \sim \frac{1}{4}\mathcal{N}(\boldsymbol{\mu}_1, \boldsymbol{\Sigma}_1) + \frac{1}{4}\mathcal{N}(\boldsymbol{\mu}_2, \boldsymbol{\Sigma}_2) + \frac{1}{4}\mathcal{N}(\boldsymbol{\mu}_3, \boldsymbol{\Sigma}_3) + \frac{1}{4}\mathcal{N}(\boldsymbol{\mu}_4, \boldsymbol{\Sigma}_4)$, where $\boldsymbol{\mu}_1 = [0, 1.5]^\top$, $\boldsymbol{\mu}_2 = [-1.5, 0]^\top$, $\boldsymbol{\mu}_3 = [1.5, 0]^\top$, $\boldsymbol{\mu}_4 = [0, -1.5]^\top$, $\boldsymbol{\Sigma}_1 = \boldsymbol{\Sigma}_4 = \left(\begin{smallmatrix} 3 & 0 \\ 0 & 0.025 \end{smallmatrix}\right)$ and $\boldsymbol{\Sigma}_2 = \boldsymbol{\Sigma}_3 = \left(\begin{smallmatrix} 0.025 & 0 \\ 0 & 3 \end{smallmatrix}\right)$. We generate 5000 $(x, y)$ pairs for each mixture component. In order to implement $\Delta$-GAN and Triple GAN-s, we model $p_x(x|y)$ and $p_y(y|x)$ as $p_x(x|y) = \int \delta(x - G_x(y, \boldsymbol{z}))p(\boldsymbol{z})d\boldsymbol{z}$, $p_y(y|x) = \int \delta(y - G_y(x, \boldsymbol{z}))p(\boldsymbol{z})d\boldsymbol{z}$ where both $G_x$ and $G_y$ are modeled as a 4-hidden-layer multilayer perceptron (MLP) with 500 hidden units in each layer. $p(\boldsymbol{z})$ is a bivariate standard Gaussian distribution. Triple GAN can be implemented by specifying both $p_x(x|y)$ and $p_y(y|x)$ to be distributions with explicit density form, *e.g.*, Gaussian distributions. However, the performance can be bad since it fails to capture the multi-modality of $p_x(x|y)$ and $p_y(y|x)$. Hence, only Triple GAN-s is implemented.

Results are shown in Figure 2. The joint distributions $p_x(x, y)$ and $p_y(x, y)$ learned by $\Delta$-GAN successfully match the true joint distribution $p(x, y)$. Triple GAN-s cannot achieve this, and can only guarantee $\frac{1}{2}(p_x(x, y) + p_y(x, y))$ matches $p(x, y)$. Although this experiment is limited due to its simplicity, the results clearly support the advantage of our proposed model over Triple GAN.

## 4.2 Semi-supervised classification

We evaluate semi-supervised classification on the CIFAR10 dataset with 4000 labels. The labeled data is distributed equally across classes and the results are averaged over 10 runs with different random splits of the training data. For fair comparison, we follow the publically available code of Triple GAN and use the same regularization terms and hyperparameter settings as theirs. Results are summarized in Table 1. Our $\Delta$-GAN achieves the best performance among all the competing methods. We also show the ability of $\Delta$-GAN to disentangle classes and styles in Figure 3. $\Delta$-GAN can generate realistic data in a specific class and the injected noise vector encodes meaningful style patterns like background and color.

## 4.3 Image-to-image translation

We first evaluate image-to-image translation on the edges2shoes dataset. Results are shown in Figure 4(bottom). Though DiscoGAN is an unsupervised learning method, it achieves impressive results. However, with supervision provided by 10% paired data, $\Delta$-GAN generally generates more accurate edge details of the shoes. In order to provide quantitative evaluation of translating shoes to edges, we use mean squared error (MSE) as our metric. The MSE of using DiscoGAN is 140.1; with 10%, 20%, 100% paired data, the MSE of using $\Delta$-GAN is 125.3, 113.0 and 66.4, respectively.

To further demonstrate the importance of providing supervision of domain correspondence, we created a new dataset based on MNIST [34], where the two image domains are the MNIST images and their corresponding tranposed ones. As can be seen in Figure 4(top), $\Delta$-GAN matches images

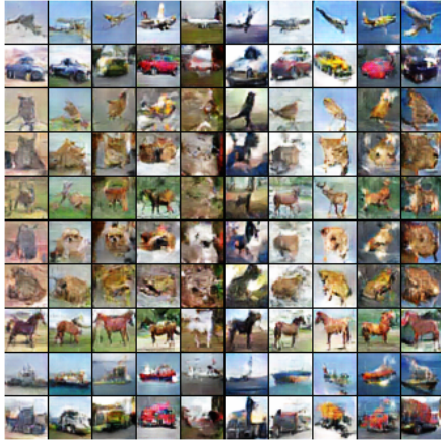

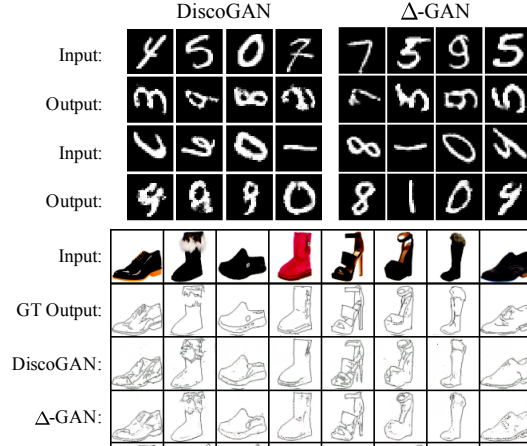

Figure 3: Generated CIFAR10 samples, where each row shares the same label and each column uses the same noise.

Figure 4: Image-to-image translation experiments on the MNIST-to-MNIST-transpose and edges2shoes datasets.

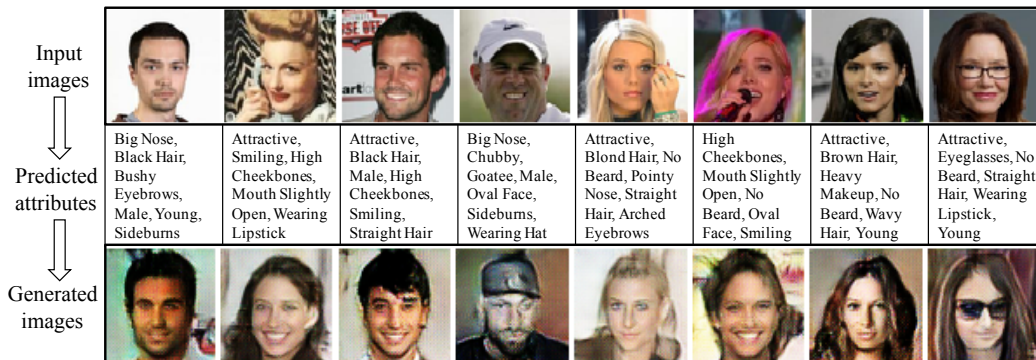

Figure 5: Results on the face-to-attribute-to-face experiment. The 1st row is the input images; the 2nd row is the predicted attributes given the input images; the 3rd row is the generated images given the predicted attributes.

Table 3: Results of P@10 and nDCG@10 for attribute predicting on CelebA and COCO.

| Dataset | CelebA | | | COCO | | |
|---|---|---|---|---|---|---|
| Method | 1% | 10% | 100% | 10% | 50% | 100% |
| Triple GAN | 40.97/50.74 | 62.13/73.56 | 70.12/79.37 | 32.64/35.91 | 34.00/37.76 | 35.35/39.60 |
| Δ-GAN | **53.21/58.39** | **63.68/75.22** | **70.37/81.47** | **34.38/37.91** | **36.72/40.39** | **39.05/42.86** |

betwen domains well, while DiscoGAN fails in this task. For supporting quantitative evaluation, we have trained a classifier on the MNIST dataset, and the classification accuracy of this classifier on the test set approaches 99.4%, and is, therefore, trustworthy as an evaluation metric. Given an input MNIST image $x$, we first generate a transposed image $y$ using the learned generator, and then manually transpose it back to normal digits $y^T$, and finally send this new image $y^T$ to the classifier. Results are summarized in Table 2, which are averages over 5 runs with different random splits of the training data. Δ-GAN achieves significantly better performance than Triple GAN and DiscoGAN.

## 4.4 Attribute-conditional image generation

We apply our method to face images from the CelebA dataset. This dataset consists of 202,599 images annotated with 40 binary attributes. We scale and crop the images to $64 \times 64$ pixels. In order to qualitatively evaluate the learned attribute-conditional image generator and the multi-label classifier, given an input face image, we first use the classifier to predict attributes, and then use the image generator to produce images based on the predicted attributes. Figure 5 shows example results. Both the learned attribute predictor and the image generator provides good results. We further show another set of image editing experiment in Figure 6. For each subfigure, we use a same set of attributes with different noise vectors to generate images. For example, for the top-right subfigure,

1st row + *pale skin* = 2nd row     1st row + *eyeglasses* = 2nd row

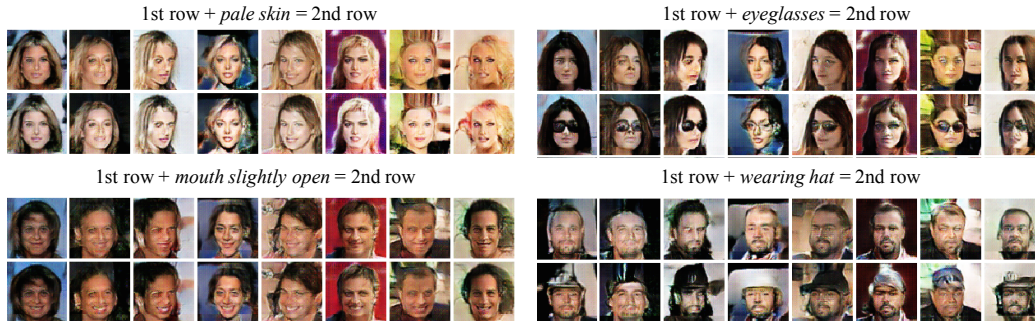

1st row + *mouth slightly open* = 2nd row     1st row + *wearing hat* = 2nd row

Figure 6: Results on the image editing experiment.

| Input | Predicted attributes | Generated images | Input | Predicted attributes | Generated images |
|---|---|---|---|---|---|
| | baseball, standing, next, player, man, group, person, field, sport, ball, outdoor, game, grass, crowd | | | tennis, player, court, man, playing, field, racket, sport, swinging, ball, outdoor, holding, game, grass | |
| | surfing, people, woman, water, standing, wave, man, top, riding, sport, ocean, outdoor, board | | | skiing, man, group, covered, day, hill, person, snow, riding, outdoor | |
| | red, sign, street, next, pole, outdoor, stop, grass | | | pizza, rack, blue, grill, plate, stove, table, pan, holding, pepperoni, cooked | |
| | sink, shower, indoor, tub, restroom, bathroom, small, standing, room, tile, white, stall, tiled, black, bath | | | computer, laptop, room, front, living, indoor, table, desk | |

Figure 7: Results on the image-to-attribute-to-image experiment.

all the images in the 1st row were generated based on the following attributes: *black hair, female, attractive*, and we then added the attribute of "*sunglasses*" when generating the images in the 2nd row. It is interesting to see that $\Delta$-GAN has great flexibility to adjust the generated images by changing certain input attribtutes. For instance, by switching on the *wearing hat* attribute, one can edit the face image to have a hat on the head.

In order to demonstrate the scalablility of our model to large and complex datasets, we also present results on the COCO dataset. Following [37], we first select a set of 1000 attributes from the caption text in the training set, which includes the most frequent nouns, verbs, or adjectives. The images in COCO are scaled and cropped to have $64 \times 64$ pixels. Unlike the case of CelebA face images, the networks need to learn how to handle multiple objects and diverse backgrounds. Results are provided in Figure 7. We can generate reasonably good images based on the predicted attributes. The input and generated images also clearly share a same set of attributes. We also observe diversity in the samples by simply drawing multple noise vectors and using the same predicted attributes.

Precision (P) and normalized Discounted Cumulative Gain (nDCG) are two popular evaluation metrics for multi-label classification problems. Table 3 provides the quantatitive results of P@10 and nDCG@10 on CelebA and COCO, where @$k$ means at rank $k$ (see the Appendix for definitions). For fair comparison, we use the same network architecures for both Triple GAN and $\Delta$-GAN. $\Delta$-GAN consistently provides better results than Triple GAN. On the COCO dataset, our semi-supervised learning approach with 50% labeled data achieves better performance than the results of Triple GAN using the full dataset, demonstrating the effectiveness of our approach for semi-supervised joint distribution matching. More results for the above experiments are provided in the Appendix.

## 5  Conclusion

We have presented the Triangle Generative Adversarial Network ($\Delta$-GAN), a new GAN framework that can be used for semi-supervised joint distribution matching. Our approach learns the bidirectional mappings between two domains with a few paired samples. We have demonstrated that $\Delta$-GAN may be employed for a wide range of applications. One possible future direction is to combine $\Delta$-GAN with sequence GAN [38] or textGAN [23] to model the joint distribution of image-caption pairs.

**Acknowledgements**   This research was supported in part by ARO, DARPA, DOE, NGA and ONR.

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
