[Supplementary Material]

# Triangle Generative Adversarial Networks: Supplementary Material

**Zhe Gan**[*], **Liqun Chen**[*], **Weiyao Wang, Yunchen Pu, Yizhe Zhang,**
**Hao Liu, Chunyuan Li, Lawrence Carin**
Duke University
zhe.gan@duke.edu

## A    Detailed theoretical analysis

**Proposition 1.** *For any fixed generator $G_x$ and $G_y$, the optimal discriminator $D_1$ and $D_2$ of the game defined by the value function $V(G_x, G_y, D_1, D_2)$ is*

$$D_1^*(\boldsymbol{x}, \boldsymbol{y}) = \frac{p(\boldsymbol{x}, \boldsymbol{y})}{p(\boldsymbol{x}, \boldsymbol{y}) + p_x(\boldsymbol{x}, \boldsymbol{y}) + p_y(\boldsymbol{x}, \boldsymbol{y})}, \ \ D_2^*(\boldsymbol{x}, \boldsymbol{y}) = \frac{p_x(\boldsymbol{x}, \boldsymbol{y})}{p_x(\boldsymbol{x}, \boldsymbol{y}) + p_y(\boldsymbol{x}, \boldsymbol{y})}\,.$$

*Proof.* The training criterion for the discriminator $D_1$ and $D_2$, given any generator $G_x$ and $G_y$, is to maximize the quantity $V(G_x, G_y, D_1, D_2)$:

$$\begin{aligned}
V(G_x, G_y, D_1, D_2) = &\int_{\boldsymbol{x}} \int_{\boldsymbol{y}} p(\boldsymbol{x}, \boldsymbol{y}) \log D_1(\boldsymbol{x}, \boldsymbol{y}) d\boldsymbol{x} d\boldsymbol{y} + \int_{\boldsymbol{x}} \int_{\boldsymbol{y}} p_x(\boldsymbol{x}, \boldsymbol{y}) \log(1 - D_1(\boldsymbol{x}, \boldsymbol{y})) d\boldsymbol{x} d\boldsymbol{y} \\
&+ \int_{\boldsymbol{x}} \int_{\boldsymbol{y}} p_x(\boldsymbol{x}, \boldsymbol{y}) \log D_2(\boldsymbol{x}, \boldsymbol{y}) d\boldsymbol{x} d\boldsymbol{y} + \int_{\boldsymbol{x}} \int_{\boldsymbol{y}} p_y(\boldsymbol{x}, \boldsymbol{y}) \log(1 - D_1(\boldsymbol{x}, \boldsymbol{y})) d\boldsymbol{x} d\boldsymbol{y} \\
&+ \int_{\boldsymbol{x}} \int_{\boldsymbol{y}} p_y(\boldsymbol{x}, \boldsymbol{y}) \log(1 - D_2(\boldsymbol{x}, \boldsymbol{y})) d\boldsymbol{x} d\boldsymbol{y}\,.
\end{aligned}$$

Following [1], for any $(a, b) \in \mathbb{R}^2 \backslash \{0, 0\}$, the function $y \to a \log y + b \log(1 - y)$ achieves its maximum in $[0, 1]$ at $\frac{a}{a+b}$. This concludes the proof.    $\square$

**Proposition 2.** *The equilibrium of $V(G_x, G_y, D_1, D_2)$ is achieved if and only if $p(\boldsymbol{x}, \boldsymbol{y}) = p_x(\boldsymbol{x}, \boldsymbol{y}) = p_y(\boldsymbol{x}, \boldsymbol{y})$ with $D_1^*(\boldsymbol{x}, \boldsymbol{y}) = \frac{1}{3}$ and $D_2^*(\boldsymbol{x}, \boldsymbol{y}) = \frac{1}{2}$, and the optimum value is $-3 \log 3$.*

*Proof.* Given the optimal $D_1^*(\boldsymbol{x}, \boldsymbol{y})$ and $D_2^*(\boldsymbol{x}, \boldsymbol{y})$, the minimax game can be reformulated as:

$$C(G_x, G_y) = \max_{D_1, D_2} V(G_x, G_y, D_1, D_2) \tag{1}$$

$$= \mathbb{E}_{(\boldsymbol{x}, \boldsymbol{y}) \sim p(\boldsymbol{x}, \boldsymbol{y})} \left[ \log \frac{p(\boldsymbol{x}, \boldsymbol{y})}{p(\boldsymbol{x}, \boldsymbol{y}) + p_x(\boldsymbol{x}, \boldsymbol{y}) + p_y(\boldsymbol{x}, \boldsymbol{y})} \right] \tag{2}$$

$$+ \mathbb{E}_{(\boldsymbol{x}, \boldsymbol{y}) \sim p_x(\boldsymbol{x}, \boldsymbol{y})} \left[ \log \frac{p_x(\boldsymbol{x}, \boldsymbol{y})}{p(\boldsymbol{x}, \boldsymbol{y}) + p_x(\boldsymbol{x}, \boldsymbol{y}) + p_y(\boldsymbol{x}, \boldsymbol{y})} \right] \tag{3}$$

$$+ \mathbb{E}_{(\boldsymbol{x}, \boldsymbol{y}) \sim p_y(\boldsymbol{x}, \boldsymbol{y})} \left[ \log \frac{p_y(\boldsymbol{x}, \boldsymbol{y})}{p(\boldsymbol{x}, \boldsymbol{y}) + p_x(\boldsymbol{x}, \boldsymbol{y}) + p_y(\boldsymbol{x}, \boldsymbol{y})} \right]\,. \tag{4}$$

---

[*] Equal contribution.

Note that

$$C(G_1, G_2) = -3\log 3 + KL\Big(p(\boldsymbol{x}, \boldsymbol{y})\Big|\Big|\frac{p(\boldsymbol{x}, \boldsymbol{y}) + p_x(\boldsymbol{x}, \boldsymbol{y}) + p_y(\boldsymbol{x}, \boldsymbol{y})}{3}\Big) \tag{5}$$

$$+ KL\Big(p_x(\boldsymbol{x}, \boldsymbol{y})\Big|\Big|\frac{p(\boldsymbol{x}, \boldsymbol{y}) + p_x(\boldsymbol{x}, \boldsymbol{y}) + p_y(\boldsymbol{x}, \boldsymbol{y})}{3}\Big) \tag{6}$$

$$+ KL\Big(p_y(\boldsymbol{x}, \boldsymbol{y})\Big|\Big|\frac{p(\boldsymbol{x}, \boldsymbol{y}) + p_x(\boldsymbol{x}, \boldsymbol{y}) + p_y(\boldsymbol{x}, \boldsymbol{y})}{3}\Big). \tag{7}$$

Therefore,

$$C(G_1, G_2) = -3\log 3 + 3 \cdot JSD\Big(p(\boldsymbol{x}, \boldsymbol{y}), p_x(\boldsymbol{x}, \boldsymbol{y}), p_y(\boldsymbol{x}, \boldsymbol{y})\Big) \geq -3\log 3, \tag{8}$$

where $JSD_{\pi_1, \ldots, \pi_n}(p_1, p_2, \ldots, p_n) = H\Big(\sum_{i=1}^n \pi_i p_i\Big) - \sum_{i=1}^n \pi_i H(p_i)$ is the Jensen-Shannon divergence. $\pi_1, \ldots, \pi_n$ are weights that are selected for the probability distribution $p_1, p_2, \ldots, p_n$, and $H(p)$ is the entropy for distribution $p$. In the three-distribution case described above, we set $n = 3$ and $\pi_1 = \pi_2 = \pi_3 = \frac{1}{3}$.

For $p(\boldsymbol{x}, \boldsymbol{y}) = p_x(\boldsymbol{x}, \boldsymbol{y}) = p_y(\boldsymbol{x}, \boldsymbol{y})$, we have $D_1^*(\boldsymbol{x}, \boldsymbol{y}) = \frac{1}{3}$, $D_2^*(\boldsymbol{x}, \boldsymbol{y}) = \frac{1}{2}$ and $C(G_x, G_y) = -3\log 3$. Since the Jensen-Shannon divergence is always non-negative, and zero iff they are equal, we have shown that $C^* = -3\log 3$ is the global minimum of $C(G_x, G_y)$ and that the only solution is $p(\boldsymbol{x}, \boldsymbol{y}) = p_x(\boldsymbol{x}, \boldsymbol{y}) = p_y(\boldsymbol{x}, \boldsymbol{y})$, *i.e.*, the generative models perfectly replicating the data distribution. $\square$

## B $\Delta$-GAN training procedure

---
**Algorithm 1** $\Delta$-GAN training procedure.

---
$\boldsymbol{\theta}_g, \boldsymbol{\theta}_d \leftarrow$ initialize network parameters
**repeat**
   $(\boldsymbol{x}_p^{(1)}, \boldsymbol{y}_p^{(1)}), \ldots, (\boldsymbol{x}_p^{(M)}, \boldsymbol{y}_p^{(M)}) \sim p(\boldsymbol{x}, \boldsymbol{y})$                     $\triangleright$ Get $M$ paired data samples
   $\boldsymbol{x}_u^{(1)}, \ldots, \boldsymbol{x}_u^{(M)} \sim p(\boldsymbol{x})$                                $\triangleright$ Get $M$ unpaired data samples
   $\boldsymbol{y}_u^{(1)}, \ldots, \boldsymbol{y}_u^{(M)} \sim p(\boldsymbol{y})$
   $\tilde{\boldsymbol{x}}_u^{(i)} \sim p_x(\boldsymbol{x}|\boldsymbol{y} = \boldsymbol{y}_u^{(i)}), \quad i = 1, \ldots, M$            $\triangleright$ Sample from the conditionals
   $\tilde{\boldsymbol{y}}_u^{(j)} \sim p_y(\boldsymbol{y}|\boldsymbol{x} = \boldsymbol{x}_u^{(j)}), \quad j = 1, \ldots, M$
   $\rho_{11}^{(i)} \leftarrow D_1(\boldsymbol{x}_p^{(i)}, \boldsymbol{y}_p^{(i)}), \quad i = 1, \ldots, M$            $\triangleright$ Compute discriminator predictions
   $\rho_{12}^{(i)} \leftarrow D_1(\tilde{\boldsymbol{x}}_u^{(i)}, \boldsymbol{y}_u^{(i)}), \rho_{13}^{(i)} \leftarrow D_1(\boldsymbol{x}_u^{(i)}, \tilde{\boldsymbol{y}}_u^{(i)}), \quad i = 1, \ldots, M$
   $\rho_{21}^{(i)} \leftarrow D_2(\tilde{\boldsymbol{x}}_u^{(i)}, \boldsymbol{y}_u^{(i)}), \rho_{22}^{(i)} \leftarrow D_2(\boldsymbol{x}_u^{(i)}, \tilde{\boldsymbol{y}}_u^{(i)}), \quad i = 1, \ldots, M$
   $\mathcal{L}_{d_1} \leftarrow -\frac{1}{M}\sum_{i=1}^M \log\rho_{11}^{(i)} - \frac{1}{M}\sum_{j=1}^M \log(1 - \rho_{12}^{(j)}) - \frac{1}{M}\sum_{k=1}^M \log(1 - \rho_{13}^{(k)})$
   $\mathcal{L}_{d_2} \leftarrow -\frac{1}{M}\sum_{i=1}^M \log\rho_{21}^{(i)} - \frac{1}{M}\sum_{j=1}^M \log(1 - \rho_{22}^{(j)})$       $\triangleright$ Compute discriminator loss
   $\mathcal{L}_{g_1} \leftarrow -\frac{1}{M}\sum_{i=1}^M \log\rho_{12}^{(i)} - \frac{1}{M}\sum_{j=1}^M \log(1 - \rho_{21}^{(j)})$          $\triangleright$ Compute generator loss
   $\mathcal{L}_{g_2} \leftarrow -\frac{1}{M}\sum_{i=1}^M \log\rho_{13}^{(i)} - \frac{1}{M}\sum_{j=1}^M \log\rho_{22}^{(j)}$
   $\boldsymbol{\theta}_d \leftarrow \boldsymbol{\theta}_d - \nabla_{\boldsymbol{\theta}_d}(\mathcal{L}_{d_1} + \mathcal{L}_{d_2})$        $\triangleright$ Gradient update on discriminator networks
   $\boldsymbol{\theta}_g \leftarrow \boldsymbol{\theta}_g - \nabla_{\boldsymbol{\theta}_g}(\mathcal{L}_{g_1} + \mathcal{L}_{g_2})$          $\triangleright$ Gradient update on generator networks
**until** convergence

---

## C Additional experimental results

Figure 1: Additional results on the image-to-image translation experiment.

Figure 2: Additional results on the face-to-attribute-to-face experiment.

Figure 3: Additional results on the image editing experiment.

Figure 4: Additional results on the image-to-attribute-to-image experiment.

Figure 5: Attribute-conditional image generation on the COCO dataset. Input attributes are omited for brevity.

Table 1: Results of P@5 and nDCG@5 for attribute predicting on CelebA and COCO.

| Dataset | CelebA | | | COCO | | |
|---|---|---|---|---|---|---|
| Method | 1% | 10% | 100% | 10% | 50% | 100% |
| Triple GAN | 49.35/52.73 | 73.68/74.55 | 80.89/78.58 | 38.98/41.00 | 41.08/43.50 | 43.18/46.00 |
| $\Delta$-GAN | 59.55/60.53 | 74.06/75.49 | 80.39/79.41 | 41.51/43.55 | 44.42/46.40 | 47.32/49.24 |

Table 2: Results of P@3 and nDCG@3 for attribute predicting on CelebA and COCO.

| Dataset | CelebA | | | COCO | | |
|---|---|---|---|---|---|---|
| Method | 1% | 10% | 100% | 10% | 50% | 100% |
| Triple GAN | 55.30/56.87 | 76.09/75.44 | 83.54/84.74 | 42.23/43.60 | 45.35/46.85 | 48.47/50.10 |
| $\Delta$-GAN | 62.62/62.72 | 76.04/76.27 | 84.81/86.85 | 45.45/46.56 | 48.19/49.29 | 50.92/52.02 |

## D  Evaluation metrics for multi-label classification

**Precision@$k$**   Precision at $k$ is a popular evaluation metric for multi-label classification problems. Given the ground truth label vector $\boldsymbol{y} \in \{0,1\}^L$ and the prediction $\hat{\boldsymbol{y}} \in [0,1]^L$, P@$k$ is defined as

$$P@k := \frac{1}{k} \sum_{l \in \text{rank}_k(\hat{\boldsymbol{y}})} y^{(l)} \, .$$

Precision at $k$ performs evaluation that counts the fraction of correct predictions in the top $k$ scoring labels.

**nDCG@$k$**   normalized Discounted Cumulative Gain (nDCG) at rank $k$ is a family of ranking measures widely used in multi-label learning. DCG is the total gain accumulated at a particular rank $p$, which is defined as

$$DCG@k := \sum_{l \in \text{rank}_k(\hat{\boldsymbol{y}})} \frac{y^{(l)}}{\log(l+1)} \, .$$

Then normalizing DCG by the value at rank $k$ of the ideal ranking gives

$$N@k := \frac{DCG@k}{\sum_{l=1}^{\min(k, \|\boldsymbol{y}\|_0)} \frac{1}{\log(l+1)}} \, .$$

# E Detailed network architectures

For the CIFAR10 dataset, we use the same network architecture as used in Triple GAN [2]. For the edges2shoes dataset, we use the same network architecture as used in the pix2pix paper [3]. For other datasets, we provide the detailed network architectures below.

Table 3: Architecture of the models for $\Delta$-GAN on MNIST. BN denotes batch normalization.

| Generator A to B | Generator B to A | Discriminator |
|---|---|---|
| Input $28 \times 28$ Gray Image | Input $28 \times 28$ Gray Image | Input two $28 \times 28$ Gray Image |
| $5 \times 5$ conv. 32 ReLU, stride 2, BN | $5 \times 5$ conv. 32 ReLU, stride 2, BN | $5 \times 5$ conv. 32 ReLU, stride 2, BN |
| $5 \times 5$ conv. 64 ReLU, stride 2, BN | $5 \times 5$ conv. 64 ReLU, stride 2, BN | $5 \times 5$ conv. 64 ReLU, stride 2, BN |
| $5 \times 5$ conv. 128 ReLU, stride 2, BN | $5 \times 5$ conv. 128 ReLU, stride 2, BN | $5 \times 5$ conv. 128 ReLU, stride 2, BN |
| Dropout: 0.1 | Dropout: 0.1 | Dropout: 0.1 |
| MLP output $28 \times 28$, sigmoid | MLP output $28 \times 28$, sigmoid | MLP output 1, sigmoid |

Table 4: Architecture of the models for $\Delta$-GAN on CelebA. BN denotes batch normalization. lReLU denotes Leaky ReLU.

| Generator A to B | Generator B to A | Discriminator |
|---|---|---|
| Input $64 \times 64 \times 3$ Image | Input $1 \times 40$ attributes, $1 \times 100$ noise | Input $64 \times 64$ Image and $1 \times 40$ attributes |
| $4 \times 4$ conv. 32 lReLU, stride 2, BN | concat input | concat two inputs |
| $4 \times 4$ conv. 64 lReLU, stride 2, BN | MLP output 1024, lReLU, BN | $5 \times 5$ conv. 64 ReLU, stride 2, BN |
| $4 \times 4$ conv. 128 lReLU, stride 2, BN | MP output 8192, lReLU, BN | $5 \times 5$ conv. 128 ReLU, stride 2, BN |
| $4 \times 4$ conv. 256 lReLU, stride 2, BN | concat attributes | |
| $4 \times 4$ conv. 512 lReLU, stride 2, BN | $5 \times 5$ deconv. 256 ReLU, stride 2, BN | $5 \times 5$ conv. 256 ReLU, stride 2, B |
| MLP output 512, lReLU | $5 \times 5$ deconv. 128 ReLU, stride 2, BN | $5 \times 5$ conv. 512 ReLU, stride 2, BN |
| MLP output 40, sigmoid | $5 \times 5$ deconv. 64 ReLU, stride 2, BN | |
| | $5 \times 5$ deconv. 3 tanh, stride 2, BN | MLP output 1, sigmoid |

Table 5: Architecture of the models for $\Delta$-GAN on COCO. BN denotes batch normalization. lReLU denotes Leaky ReLU. $Dim$ denotes the number of attributes.

| Generator A to B | Generator B to A | Discriminator |
|---|---|---|
| Input $64 \times 64 \times 3$ Image | Input $1 \times 40$ attributes, $1 \times 100$ noise | Input $64 \times 64$ Image and $1 \times Dim$ attributes |
| $4 \times 4$ conv. 32 lReLU, stride 2, BN | concat inputs | concat conditional inputs |
| $4 \times 4$ conv. 64 lReLU, stride 2, BN | MLP output 16384, BN | |
| $4 \times 4$ conv. 128 lReLU, stride 2, BN | ResNet Block | $5 \times 5$ conv. 64 ReLU, stride 2, BN |
| $4 \times 4$ conv. 256 lReLU, stride 2, BN | $4 \times 4$ deconv. 512, stride 2 | |
| $4 \times 4$ conv. 512 lReLU, stride 2, BN | $3 \times 3$ conv. 512, stride 1, BN | |
| | ResNet Block | $5 \times 5$ conv. 128 ReLU, stride 2, BN |
| ResNet Block | $4 \times 4$ deconv. 256, stride 2 | |
| | $3 \times 3$ conv. 256, stride 1, BN | $5 \times 5$ conv. 256 ReLU, stride 2, BN |
| | $4 \times 4$ deconv. 128 ReLU, stride 2 | |
| $1 \times 1$ conv. 512 lReLU, stride 1, BN | $3 \times 3$ conv. 128, stride 1, BN | $5 \times 5$ conv. 512 ReLU, stride 2, BN |
| | $4 \times 4$ deconv. $Dim$, stride 2 | |
| $4 \times 4$ conv. $Dim$ sigmoid, stride 4 | $3 \times 3$ conv. $Dim$ tanh, stride 1 | MLP output 1, sigmoid |