[Reviews · NeurIPS 2017]

Reviewer 1



----- Post-rebuttal: I think R2 brought up several important points that were not completely addressed by the rebuttal. Most importantly, I agree that the characterization of Triple GAN is somewhat misleading. The current paper should clarify that Triangle GAN fits a model to p_y(y|x) rather than this density being required as given. The toy experiment should note that p_y(y|x) in Triple GAN could be modeled as a mixture of Gaussians, although it is preferable that Triangle GAN does not require specifying this. ----- This is a nice paper. The objective comes down to conditional GAN + BiGAN/ALI. That is an intuitive and perhaps simple thing to try for the semi-supervised setting, but it’s nice that this paper backs up the formulation with theory about behavior at optimality. The results look good and it gives a principled framework for semi-supervised learning with GANs. The experiments are thorough and convincing. I was impressed to see state-of-the-art semi-supervised results not just on "generative" tasks like synthesizing images, but also on CIFAR label prediction (Table 1). The paper is clearly written, appears to be technically correct, and overall well executed. I also appreciated the careful comparison to Triple GAN, with the toy experiment in Figure 2 being especially helpful. Section 4.3: it would be nice to also compare to the setting of having 0% labeled data, here and perhaps elsewhere. Here especially since DiscoGAN doesn’t use labels. The CycleGAN paper actually already includes this experiment (it reduces to just BiGAN), so that paper could also just be referred to. It’s surprising that DiscoGAN does badly on the transposed MNIST metric, since DiscoGAN is explicitly trained to minimize reconstruction error on X—>Y—>X problems, which it sounds like is what this metric is measuring. Even if DiscoGAN does terribly at X—>Y, it should still do well at X—>Y—>X, if the reconstruction cost is set high enough. Therefore, this evaluation metric doesn’t seem the most appropriate for showing the possible advantages of the current method over DiscoGAN. Fig 4 is compelling though, since I can actually see that on X—>Y, DiscoGAN fails. Minor comments: 1. Line 33: “infinite” —> “infinitely” 2. Line 138: “strive” —> “strives” 3. Line 187: I would separate DTN from DiscoGAN, CycleGAN, and DualGAN. The latter all use an essentially identical objective, whereas the former is a bit different.

Reviewer 2



In this paper, the authors have presented the Triangle Generative Adversarial Network, a new GAN framework that can be used for semi-supervised joint distribution matching. The proposed approach learns the bidirectional mappings between two domains with a few paired samples. They have demonstrated that Triangle-GAN may be employed for a wide range of applications. I have the following concerns. (1) The training time of this new network is not reported. The authors may want to have some discussion on the training time. (2) Is this network stable for training? Or is there any trick for efficiently training this network? (3) The performance gain of Triangle GAN is not significant compared to ALI[7]. The authors made this network much more complex but failed to achieve satisfied performance improvement. (4) Fig. 3 is not well demonstrated. I understand that the proposed network can be used to generate samples. However, the improvement is not well explained.

Reviewer 3



Triangle GAN is an elegant method to perform translation across domains with limited numbers of paired training samples. Empirical results seem good, particularly the disentanglement of known attributes and noise vectors. I'm worried, though, that the comparison to the closest similar method is somewhat unfair, and in general that the relationship to it is somewhat unclear. Relation to Triple GAN: Your statements about the Triple GAN model's requirements on p_y(y | x) are somewhat misleading. Both in lines 53-56 and section 2.5, it sounds like you're saying that p_y(y | x) must be known a priori, which would be a severe limitation to their model. This, though, is not the case: rather, the Triple GAN model learns a classifier which estimates p_y(y | x), and show (their Theorem 3.3) that this still leads to the unique desired equilibrium. In cases where p_y(y | x) does not have a density, it is true that their loss function would be undefined. But their algorithm will still simply estimate a classifier which gives high probability densities, and I think their Theorem 3.3 still applies; the equilibrium will simply be unreachable. Your Propositions 1 and 2 also break when the distributions lack a joint density, and so this is not a real advantage of your method. The asymmetry in the Triple GAN loss function (8), however, still seems potentially undesirable, and your approach seems preferable in that respect. But depending on the complexity of the domains, it may be easier or harder to define and learn a classifier than a conditional generator, and so this relationship is not totally clear. In fact, it seems like your image-label mappings (lines 164-173) use essentially this approach of a classifier rather than a conditional generator as you described earlier. This model appears sufficiently different from your main Triangle GAN approach that burying its description in a subsection titled Applications is strange. More discussion about its relationship to Triangle GAN, and why the default approach didn't work for you here, Similarly, the claim that Triple GAN cannot be implemented on your toy data experiment (section 4.1) seems completely wrong: it perhaps cannot with exactly the modeling choice you make of using G_x and G_y, but p_x(x | y) and p_y(y | x) are simple Gaussians and can certainly be modeled with e.g. a network outputting a predictive mean and variance, which is presumably what the Triple GAN approach would be in this case. Theory: The claim that "p(x, y) = p_x(x, y) = p_y(x, y) can be achieved in theory" (line 120) is iffy at best. See the seminal paper Arjovsky and Bottou. Towards Principled Methods for Training Generative Adversarial Networks. ICLR 2017. arXiv: 1701.04862. The basic arguments of this paper should apply to your model as well, rendering your theoretical analysis kind of moot. This subarea of GANs is fast-moving, but really I think any GAN work at this point, especially papers attempting to show any kind of theoretical results like your section 2.3, needs to grapple with its relationship to the results of that paper and the followup Wasserstein GAN (your [22]) line of work. As you mention, it seems that many of these issues could be addressed in your framework without too much work, but it would be useful to discuss this in the theory section rather than simply stating results that are known to be of very limited utility. Section 4.4: In Figure 6, the edited images are clearly not random samples from the training set: for example, the images where you add eyeglasses only show women. This adds a natural question of, since the images are not random, were they cherry-picked to look for good results? I would strongly recommend ensuring that all figures are based on random samples, or if that is impractical for some reason, clarifying what subset you used (and why) and being random within that subset. Smaller comments: Lines 30-33: ALI and BiGAN don't seem to really fit into the paradigm of "unsupervised joint distribution matching" very well, since in their case one of the domains is a simple reference distribution over latent codes, and *any* pairwise mapping is fine as long as it's coherent. You discuss this a little more in section 3, but it's kind of strange at first read; I would maybe put your citations of [9] to [11] first and then mention that ALI/BiGAN can be easily adapted to this case as well. Section 2.2: It would be helpful to clarify here that Triangle GANs are exclusively a "translation" method: you cannot produce an (x, y) sample de novo. This is fine for the settings considered here, but differs from typical GAN structures enough that it is probably worth re-emphasizing e.g. that when you refer to drawing a sample from p(x), you mean taking a training sample rather than some form of generative model.